# Mechanistic and microkinetic study of non-oxidative methane coupling on a single-atom iron catalyst

Seok Ki Kim [1,2 ✉], Hyun Woo Kim [3], Seung Ju Han[1], Sung Woo Lee[1], Jungho Shin[3] & Yong Tae Kim [1 ✉]

Non-oxidative methane coupling has promising economic potential, but the catalytic and radical reactions become complicated at high temperatures. Here, we investigate the mechanism of non-oxidative methane coupling on an iron single-atom catalyst using density functional theory, and evaluate the catalytic performance under various reaction conditions using microkinetic modelling and experiments. Under typical reaction conditions (1300 K and 1 bar), C–C coupling and subsequent dehydrogenation to produce ethylene shows comparable energetics between the gas-phase and catalytic pathways. However, the microkinetic analysis reveals that the iron single-atom catalyst converted methane to mainly $CH_3$ and $H_2$ at reaction temperatures above 1300 K, and acetylene production is dominant over ethylene production. The sensitivity analysis suggests that increasing the $C_2$ hydrocarbon yield by optimising the reaction conditions is limited. The experimental results obtained at 1293 K are consistent with the theoretical estimation that acetylene is the main $C_2$ product over the iron single-atom catalyst.

[1] C1 Gas & Carbon Convergent Research Center, Korea Research Institute of Chemical Technology, Daejeon 34114, Republic of Korea. [2] Advanced Materials and Chemical Engineering, University of Science and Technology, Daejeon 34113, Republic of Korea. [3] Chemical Data-Driven Research Center, Korea Research Institute of Chemical Technology, Daejeon 34114, Republic of Korea. ✉email: skkim726@krict.re.kr; ytkim@krict.re.kr

Methane constitutes the majority of various gas resources, such as natural gas, shale gas, and landfill gas. More than 500 million tons are generated each year, an amount which is gradually increasing[1]. Currently, only a small amount of methane is used as industrial feedstock relative to the total amount produced, and most is burnt as an energy source[2]. This is because methane is highly stable and requires additional oxidants such as $H_2O$ or $O_2$ to be used as a chemical feedstock, which necessitates additional equipment and operating costs[3]. If a catalyst were to be found, which could convert methane directly into chemical raw materials without the use of oxidants, the subsequent economically viable methane utilisation process could be widely commercialised[4].

Recently, a non-oxidative catalytic methane-converting technology was reported and received great attention[5]. This innovative technology succeeded in producing ethylene and benzene in high yields without coke formation at temperatures above 1200 K, where dehydrogenation is thermodynamically favourable. The previous study focused on the role of Fe-single atom catalyst that was capable of producing $CH_3$ radicals but did not participate in further C–C coupling of dehydrogenation[5]. Furthermore, a recent study on a new type of Fe single-atom catalyst for non-oxidative methane coupling reported that the Fe single sites were only active in the initial reaction period, then rapidly lost their activity and resulted in varying hydrocarbon distributions[6]. Since the C–H bonds of ethylene and benzene are more readily activated than that of methane, it is thermodynamically challenging to inhibit coke deposition at such high temperatures. Two understandings are possible: engineering the reactor such that the produced hydrocarbons rapidly exit the reaction zone before further dehydrogenation proceeds[7], or reinterpreting the role of the Fe single-atom catalyst to play a role in C–C coupling in addition to the generation of $CH_3$ radicals. The latter effect can increase the partial pressure of $C_2$-hydrocarbons and the successive trimerisation rate in the reaction stream under mild reaction conditions, where coke formation is hindered. The proposed Fe single-atom catalyst was composed of $FeC_2$ clusters embedded in a $SiO_2$ substrate[5]. Subsequent studies focusing on the reactor[7,8] and catalyst[9] were also carried out using this type of catalyst. However, experimental observations at such high reaction temperatures originate from a mixture of gas-phase radical reactions and catalysis, which are difficult to decouple.

In this study, we investigate the energetics of methane coupling by focusing on the reactions occurring on the Fe single-atom catalyst using density functional theory (DFT) calculations. Although there has not been much research on how methane reacts on the catalyst surface under non-oxidative conditions, we establish a reaction network based on the Eley–Rideal mechanism, which has been suggested as a dominant for the oxidative methane coupling reaction[10,11]. The absence of adjacent adsorption at the single-atom active site supports this assumption as well. Based on the reaction network and energetics obtained using DFT, a microkinetic model is built to analyse the activity and selectivity of the Fe single-atom catalyst under various reaction conditions. Furthermore, we design and perform a series of reactions to demonstrate the role of the Fe-single atom catalyst computationally identified.

## Results

**Reaction network and energetics.** The energetics of the C–H bond activation of methane and subsequent C–C bond formation on the FeC-confined $SiO_2$ catalyst surface were analysed. According to previous experimental and theoretical investigations[5,9], we built an initial surface geometry by substituting a $SiO_2$ unit in the (001) surface of β-tridymite with an

$FeC_2$ unit, as shown in Fig. 1a, b. Then, all possible structures that could be formed during methane decomposition and $C_2$-hydrocarbon production were generated (see Supplementary Data 1). We note that the lattice-confined C atoms can also participate in the formation of $C_2$-hydrocarbons in this model network. To exclude duplication of geometries and simplify the reaction network, we focused on the local energetics of catalysis taking place on one of the two surface –Fe–C–Si– locations. The structural notation is based on the number of C and H atoms in the local structure of interest. For instance, structures '1.2' and '3.2' indicate –Fe–$CH_2$–Si– and –Fe–C($C_2H_2$)–Si–, respectively. Variations of structures comprising the same number of atoms are distinguished with a letter, such as '2.4a' and '2.4b'. The obtained ab-initio energies of each structure were recalculated in terms of formation energies on the basis of $CH_4$ and $H_2$ and are summarised in Supplementary Data 1.

The possible reaction interconnecting structures were classified into five different categories. All the reaction categories include both forward and reverse reactions, but for concision, we denote them simply as $H_2$ abstraction, transformation, dehydrogenation, $CH_4$ insertion, and adsorption. The gaseous molecules involved in each reaction category are shown at the bottom of Fig. 1c.

Among the 108 reactions presented here, 42 transition states were randomly selected, and their formation energies were scaled with initial-state and final-state formation energies in each category, as shown in Fig. 1d and Supplementary Fig. 1a, respectively. The activation barriers for the adsorption (or desorption) of surface hydrocarbon species were calculated to be negligible and thus are not shown here. The initial-state scaling appeared to be slightly more suitable than the final-state scaling for all reaction categories. The coefficient of determination ($R^2$) values of the initial-state scaling were 0.89, 0.64, 0.97, and 0.99 for $H_2$ abstraction, transformation, dehydrogenation, and $CH_4$ insertion, respectively, whereas those of the final-state scaling were 0.72, 0.05, 0.86, and 0.79, respectively. The Brønsted–Evans–Polanyi (BEP) relation was also applied to the scaling, but this approach was found to be much weaker than both transition-state scalings, showing $R^2$ values of 0.19, 0.01, 0.87, and 0.88, respectively. Although the transition-state scaling was not necessarily stronger than the BEP relation[12], previous studies have shown that the transition-state energies of particular surface reactions, such as hydrocarbon combustion and formation at high temperatures, are more suitably estimated by transition-state scaling[13,14]. According to Hammond's postulate[15], we can speculate that the transition states are structurally more similar to the initial states than to the final states. We note that reactions in the transformation category exhibited weaker proportionality than other categories regardless of the scaling method because various types of bond cleavage and formation are combined differently for each individual transformation between surface hydrocarbon species. The initial-state scaling derived as described above for each reaction category was used to predict the remaining transition-state energies for microkinetic modelling.

To investigate the role of the catalyst in C–C coupling, a typical pathway to produce ethylene on the surface was chosen in the reaction network, and its energetics were compared with that of the gas-phase reaction pathway. We note that $CH_3$ radicals were assumed to be generated by the dehydrogenation of $CH_4$ on the catalyst surface, as illustrated in the reaction network (Fig. 1); this applies to both the catalytic and gas-phase C–C coupling pathways. In the case of catalytic C–C formation (blue pathway in Fig. 2a), methane insertion to the surface (1.0 → 2.4b), dehydrogenation to produce surface radical species (2.4b → 2.3f), $CH_3$ addition to form surface $C_2$ species (2.3f → 3.6e), and remaining dehydrogenation followed by ethylene desorption (3.6e → … → 1.0) were assumed to occur in turn. In the case of gas-phase C–C coupling (green pathway in Fig. 2a), two $CH_3$

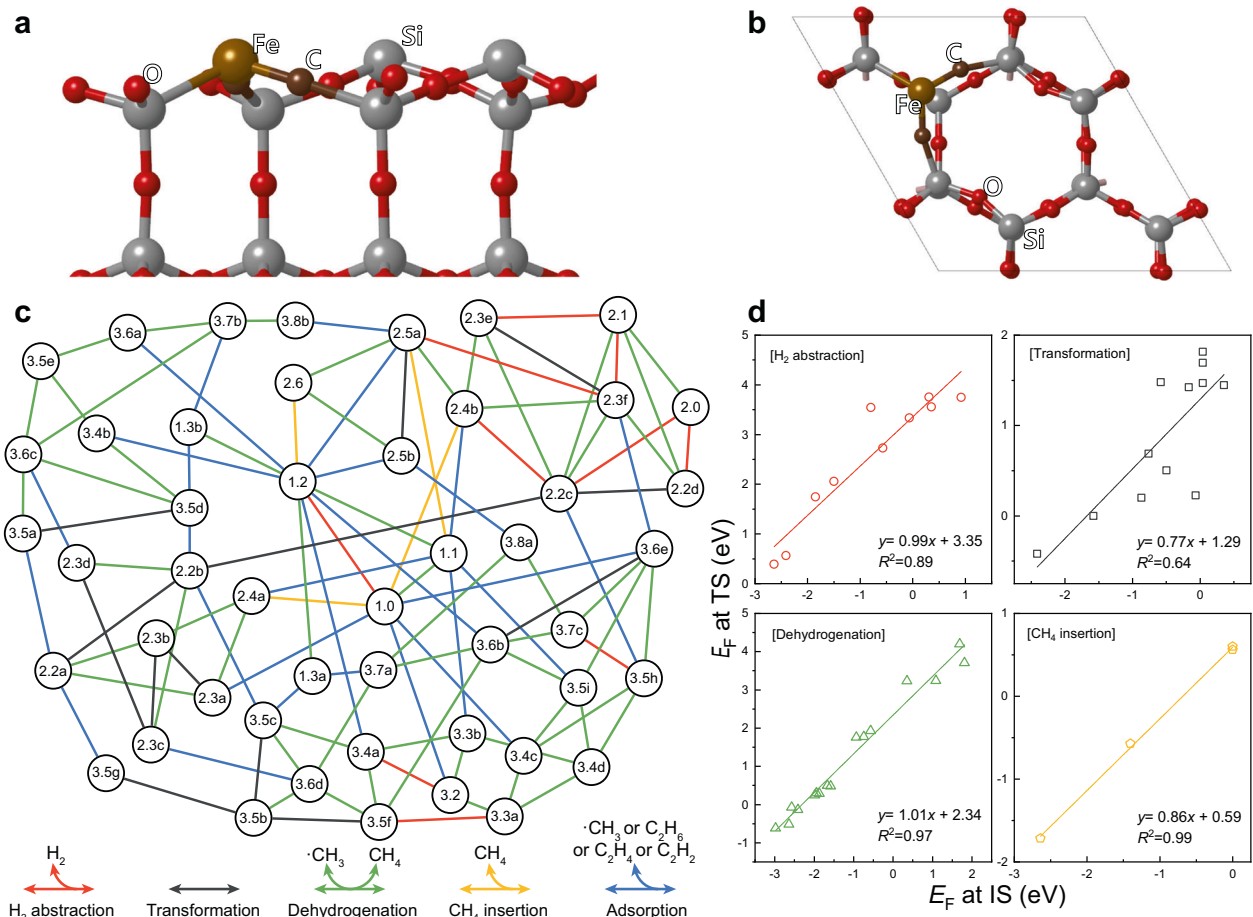

**Fig. 1 Surface structure of Fe single-atom catalyst and non-oxidative methane coupling reaction network. a, b,** Side view (**a**) and top view (**b**) of the Fe single-atom catalyst surface. An FeC$_2$ cluster was embedded in $\beta$-tridymite. **c** Non-oxidative methane coupling reaction network on the Fe single-atom catalyst. Five different reaction categories are shown below. **d** Transition-state ($E_F$ at TS) scaling with the formation energy of the initial state ($E_F$ at IS).

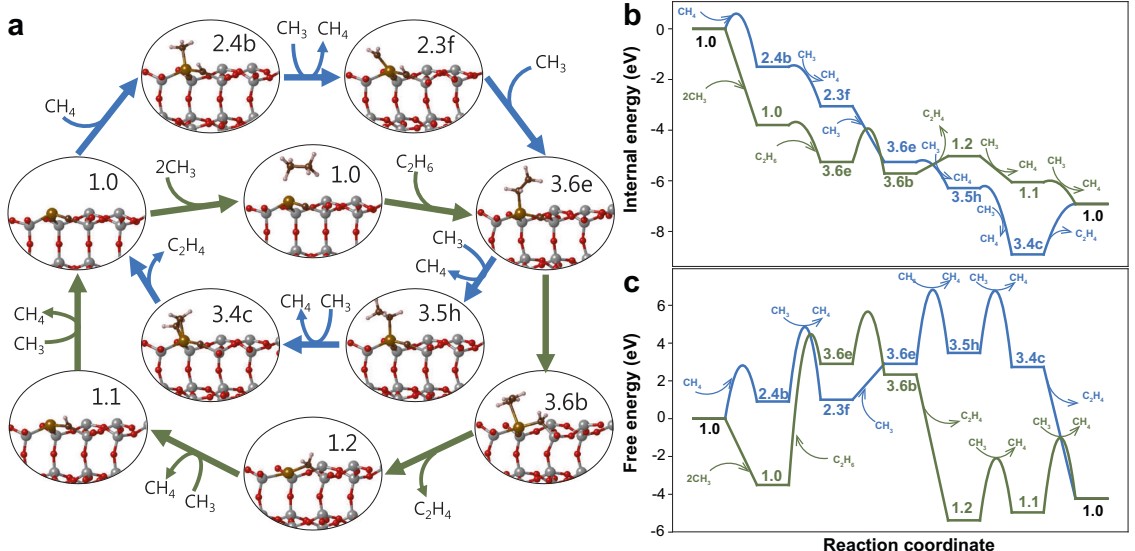

**Fig. 2 Energy comparisons between catalytic and gas-phase C–C coupling reactions. a** Configurations of surface species for catalytic (blue) and gas-phase (green) C–C couplings. **b** Internal energy comparison. **c** Free energy comparison at 1300 K and 1 bar.

radicals come together without surface mediation in the initial reaction stage ($2CH_3 \rightarrow C_2H_6$), followed by ethane insertion ($1.0 \rightarrow 3.6e$). After hydrogen transfer from the adsorbate to surface carbon ($3.6e \rightarrow 3.6b$), the surface ethylene desorbs, and surface hydrogen is removed by $\cdot CH_3$ ($1.2 \rightarrow 1.1 \rightarrow 1.0$). We note that the essential difference between the two pathways is the C–C coupling manner to form surface $C_2$ species ($3.6e$); additionally, the ethylene desorption routes (from $3.6e$ to $1.0$) were set differently for auxiliary comparisons.

Figure 2b shows potential energy profiles, where only electronic energies obtained using DFT calculations are plotted. As expected from the highly unstable nature of $CH_3$ radicals, gas-phase C–C coupling appeared to be significantly preferred over catalytic C–C coupling when comparing up to the formation of surface $C_2$ species ($3.6e$). When the same pathways were plotted in terms of free energy calculated assuming typical reaction conditions (temperature of 1300 K and partial pressures of $\cdot CH_3$ and $CH_4$ of 0.01 and 0.99 bar, respectively), the gas-phase C–C coupling was also favoured (Fig. 1c). In this case, however, the insertion of ethane for subsequent dehydrogenation to form ethylene required a significant amount of energy comparable to that required for catalytic ethylene formation.

The above results indicate that the methane consumption rate and C–C coupling path can vary depending on conditions such as the reaction temperature and partial pressures of the gas components. That is, when the reaction proceeds at a low temperature, where $\cdot CH_3$ radicals are barely present, gas-phase C–C coupling followed by catalytic dehydrogenation is more favourable than catalytic C–C coupling and subsequent dehydrogenation. In contrast, at a high temperature with a relatively high partial pressure of $\cdot CH_3$, both reactions take place simultaneously at a similar rate.

The complexity of the overall reaction, in which the relative rates of individual reactions depend on the reaction conditions, is shown in Fig. 3. Changes in the free energy of activation for typical dehydrogenation and $CH_4$ insertion are visualised as a function of $CH_4$ conversion and reaction temperature (Supplementary Fig. 2). Here, we assumed that $CH_4$ is converted to only $\cdot CH_3$ and $H_2$, and so the partial pressures of $CH_4$ and $CH_3$

varied linearly depending on $CH_4$ conversion. The increases in free-energy barriers with reaction temperature arise from negative entropies of activation, suggesting that the transition state is achieved by two reaction components in an associative manner[16]. The free energy barriers of dehydrogenation and $CH_4$ insertion cross each other at various points for both variables. This complicates analysing the mechanism by comparing pathway energetics or descriptors.

**Microkinetic analysis.** It is helpful to understand this complex reaction through plotting the overall reaction rate by considering individual reactions simultaneously. Figure 3 shows how the $CH_4$ consumption rate and production rates change with reaction temperature and relative partial pressures of $\cdot CH_3$ and $CH_4$. We note that all the reactions presented in Fig. 1c were considered in this microkinetic analysis, except for gas-phase C–C coupling, to focus on the role of the Fe single-atom catalyst. The $CH_4$ consumption rate monotonically increases with reaction temperature at a fixed $\cdot CH_3$ partial pressure. On the other hand, the maximum rate occurs when the $\cdot CH_3$ concentration varies at a fixed reaction temperature. The large quantity of $\cdot CH_3$, which is the main reaction product, in the reactant phase is expected to be detrimental in terms of thermodynamics. However, the $\cdot CH_3$ radicals also serve as an activator which removes the intermediates attached to the monoatomic active site.

As shown in Fig. 3b, the main products from the catalytic non-oxidative coupling of methane are $\cdot CH_3$ and $H_2$, whose production rate profiles are almost identical to that of the $CH_4$ consumption rate. The production rates of $C_2$-hydrocarbons were an order of magnitude less than those of $\cdot CH_3$ and $H_2$. In particular, the ethylene formation rate is negligible compared with the acetylene formation rate, indicating that acetylene is the major product from the catalytic C–C coupling reaction. The $C_2$ compounds are mainly produced at a high $\cdot CH_3$ concentration ($\log(P_{CH_3}) > -2$), which suggests that the $CH_3$ radical plays a major role in C–C coupling on the catalyst. However, when the reaction temperature is above 1100 K, the production rates of $C_2$ compounds decrease, since they are easily decomposed, and the reaction again produces $\cdot CH_3$ and $H_2$.

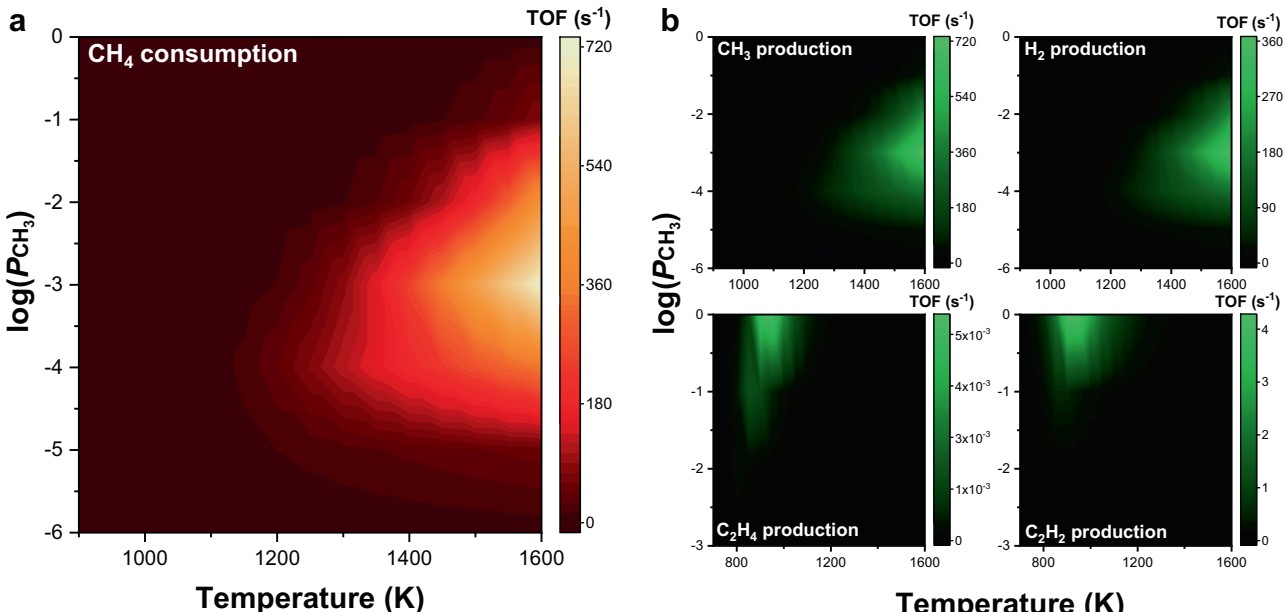

**Fig. 3 Turnover frequencies for $CH_4$ consumption and major product production. a** $CH_4$ consumption rate as a function of reaction temperature and partial pressure of $\cdot CH_3$ ($\log(P_{CH_3})$). **b** $CH_3$, $H_2$, $C_2H_4$, and $C_2H_2$ production rates as functions of reaction temperature and $\log(P_{CH_3})$.

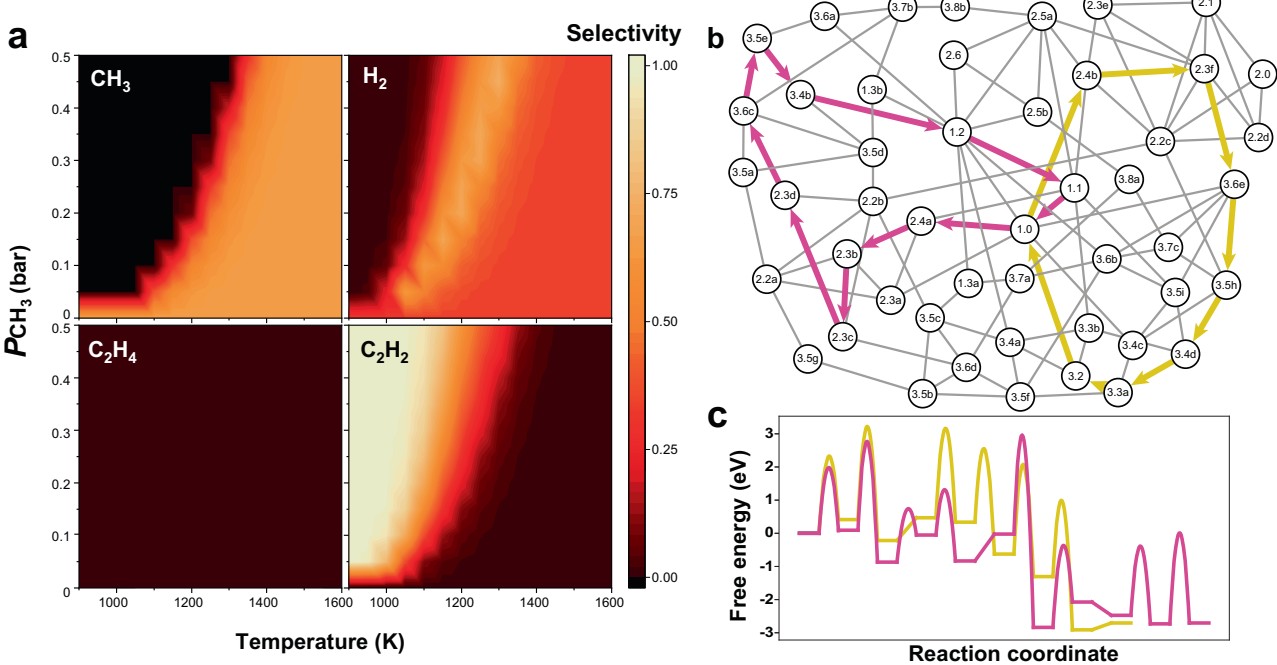

**Fig. 4 Product selectivity and dominant pathway for acetylene formation. a** $CH_3$, $H_2$, $C_2H_4$, and $C_2H_2$ selectivities as functions of reaction temperature and partial pressure of $CH_3$ ($P_{CH_3}$). **b** Two competing pathways to produce $C_2H_2$ in the reaction network, where $CH_3$ attaches to either C (intermediate 2.4a, pink lines) or Fe (intermediate 2.4b, yellow lines) on the surface. **c** Free energy diagram of the two $C_2H_2$ formation pathways at $T = 1000$ K and $P_{CH_3} = 1$ bar.

The product selectivity is shown in Fig. 4a according to changes in the reaction temperature and ·$CH_3$ concentration. As expected from the production rate (Fig. 3), $CH_4$ decomposes mainly into ·$CH_3$ and $H_2$ at temperatures above 1300 K, whereas catalytic C–C coupling to produce acetylene is dominant at and below 1200 K. We note that the ·$CH_3$ radicals also produce $CH_4$ molecules by taking surface H species in the acetylene-production region, which subsequently lowers the $CH_4$ consumption rate.

Since the major $C_2$ product was identified as acetylene, the predominant pathway for acetylene formation was traced based on individual reaction rates. Two competitive reaction pathways were found as shown in Fig. 4b. Both pathways begin with a clean surface (1.0), but one proceeds by forming intermediate 2.4a, where $CH_3$ is attached to surface carbon (2.4a, pink lines in Fig. 4b and c), whereas the other proceeds by forming intermediate 2.4b, where $CH_3$ is attached to surface Fe (2.4b, yellow lines in Fig. 4b and c). The free energy diagram (Fig. 4c) shows that both pathways have comparable reaction barriers. However, microkinetic modelling of only these pathways shows that the pathway forming 2.4b (TOF = $1.3 \times 10^{-4}$) is slightly faster than the other path (TOF = $5.9 \times 10^{-6}$). This is presumably because the former proceeds via fewer reaction steps than the latter.

The degree of rate control (DRC) was calculated as an indicator to determine how the net production rate for each product is affected by the individual reaction (Eq. 1). The higher the absolute value of sensitivity is, the greater the effect on the overall reaction rate. Figure 5 shows the DRCs of selected states for ·$CH_3$ and acetylene production with absolute values higher than 0.1. The DRCs for all the states considered in the present study are shown in Supplementary Fig. 3. If the value of a state is negative, the state must be destabilised to increase the overall reaction rate. On the contrary, if the value is positive, the overall reaction rate increases as the state stabilises. Therefore, in general, negative-valued states are strongly adsorbed species, which slow the

subsequent reaction, and positive-valued states are often transition states at the rate-determining step with a large kinetic barrier.

The production rates of ·$CH_3$ and acetylene were determined to be largely influenced by several common steps. In particular, the structure where the surface carbon is terminated with three hydrogens (1.3a, structure given in Supplementary Data 1) appears to be too stable for all reaction conditions. This is because once this structure forms, further reactions hardly proceed on the catalyst surface. On the other hand, in order to accelerate the overall reaction for both products, transition states 1.2–1.3a and 1.0–1.2 (see Supplementary Data 1 for structures) must be stabilised at high and low ·$CH_3$ partial pressures, respectively.

In the case of ·$CH_3$ production, the activation barriers of $CH_4$ insertion (1.0–2.4a and 1.0–2.4b) must be stabilised under abundant ·$CH_3$ conditions ($\log(P_{CH_3}) > -3$). However, acetylene production requires higher hydrogen transfer rates on carbon-attached structures (2.2a, 2.2b, etc.) under $CH_3$-deficient conditions ($\log(P_{CH_3}) < -3$). Discrepancies in the optimal conditions for both reactions suggest that increasing the $CH_4$ conversion and C–C coupling rate simultaneously is difficult over this catalyst.

**Reaction test**. To experimentally confirm the role of catalysis in non-oxidative methane coupling, a $SiO_2$-confined Fe catalyst (Fe@CRS, see experimental section within the Methods for the detailed preparation procedure) was prepared according to a previous report[9], and a series of experiments was carried out. In order to elucidate the structure of the Fe@CRS catalyst, an Fe K-edge X-ray absorption study was performed (Fig. 6). In the X-ray absorption near-edge spectra, the spent Fe@CRS catalyst exhibited a white line at a position similar to those of the Fe foil and $Fe_3C$ spectra, whereas the fresh catalyst showed an oxide-like spectrum. This indicates that the initial $FeO_x$ clusters in the catalyst decomposed during the reaction and transformed into

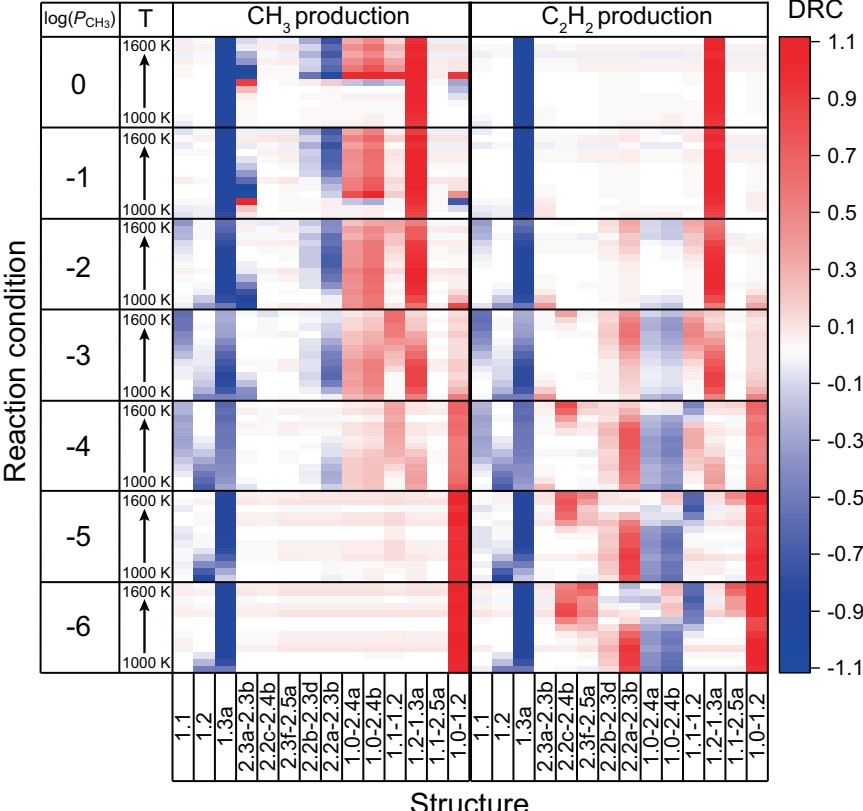

**Fig. 5 Degree of rate control calculated under different reaction conditions.** Structures with absolute DRC values greater than 0.1 are shown. DRC values for all states calculated in this study are shown in Supplementary Fig. 2.

$FeC_x$. The extended X-ray absorption fine structure of the spent Fe@CRS catalyst demonstrates that a $SiO_2$-confined Fe structure with Fe–C and Fe–Si bonds was dominantly formed[5], which validates our simulated model of the Fe single-atom surface structure.

Figure 7 shows the experimental results obtained using the Fe@CRS catalyst. Since the reaction occurs throughout both free space and on the catalyst surface in the heating zone of the reactor, we decoupled the variables into gas-phase residence time and catalyst contact time instead of using space velocity as a variable. Catalyst contact time had a great influence on increasing the $CH_4$ conversion (Fig. 7a) than gas-phase residence time, indicating that C–H activation was mainly catalysed by Fe@CRS even though gas-phase activation also contributed to some extent. Catalytic $CH_4$ conversion appears to be more active at high gas-phase residence times. This is because the radicals generated in the gas-phase reaction expose the active sites by removing adsorbed species from the catalyst surface, as predicted by the simulation.

Both the ethane and ethylene selectivities (Fig. 7b) increased with decreasing catalyst contact time, indicating that the gas-phase reaction is the major route to produce ethylene from ethane cracking. However, acetylene production appears to proceed mainly by catalysis, as the selectivity increased with catalyst contact time. These results agree with the above-discussed calculations showing that acetylene is the main $C_2$ product afforded by catalysis. Benzene selectivity increased with catalyst contact time as well as gas-phase residence time, which supports that the trimerization of acetylene occurs in the gas phase. Other hydrocarbons, including $C_3$, $C_4$, and $C_5$, were also produced during the reaction, but their total amount was as low as 20–30%

compared with that of $C_2$-hydrocarbons (see Supplementary Table 1). Coke deposition was also found to largely occur in some cases. It is noteworthy that the formation of $C_3$–$C_5$-hydrocarbons and coke increased with gas-phase residence time, indicating that the gas-phase radical reaction is considerably involved in these chain-growth reactions. Thus, further studies on the gas-phase reactions are necessary to aid in the design of catalysts and reactors.

## Discussion

In non-oxidative methane coupling, where a gas-phase radical reaction and surface catalysis are interconnected, the role of an Fe single-atom catalyst was investigated using DFT and microkinetic analyses. Based on these theoretical studies, we estimated that catalytic C–H activation occurs actively when the reaction temperature is above 1300 K, and most $CH_4$ is converted into $CH_3$ and $H_2$. At lower reaction temperatures, the catalyst also appears to mediate the C–C coupling reaction, thereby producing mainly acetylene rather than ethylene. Both the C–H activation and C–C coupling reactions on the catalyst surface were calculated to be accelerated in the presence of $CH_3$ radicals, which serve as scavengers to regenerate the catalytic active sites. However, a sensitivity analysis revealed that acetylene and ·$CH_3$ productions cannot be simultaneously promoted by changing the reaction conditions. The product distribution estimated by the computational analyses was demonstrated by a series of experiments. Experimental non-oxidative methane coupling conducted using the Fe©CRS catalyst at 1293 K showed that methane conversion and acetylene selectivity increased with increasing catalyst contact time. The results of this study are valuable towards distinguishing

the role of the catalyst in this complex reaction. However, it is necessary to consider additional gas-phase reaction kinetics to reveal the coke formation mechanism and design efficient reactors to further increase the yield of C$_2$-hydrocarbons.

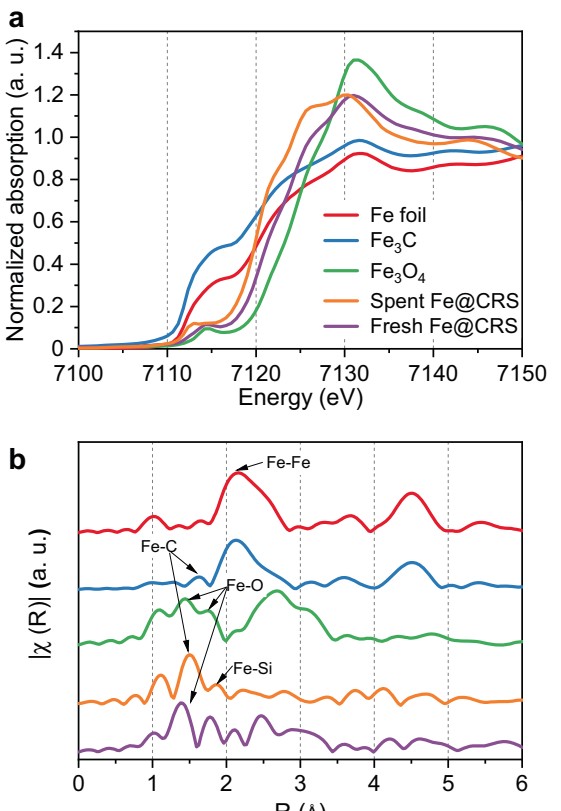

**Fig. 6 X-ray absorption spectra of the fresh and spent Fe@CRS catalysts and Fe foil, Fe$_3$C, and Fe$_3$O$_4$ reference samples. a** X-ray absorption near-edge spectra and **b** extended X-ray absorption fine structures.

## Methods

**Computational**. Electronic structure calculations were performed with DFT using the Vienna Ab-Initio Simulation Package[17,18] implementing a plane-wave basis set. The vdW-DF2 exchange-correlation functional[19,20] was used to take dispersion interactions into account within a generalised gradient approximation. The cut-off energy of the plane-wave basis was set to 450 eV, and the convergence criterion for the total energy between electronic steps was set to $1 \times 10^{-4}$ eV.

The (001) surface of β-tridymite (SiO$_2$, P6$_3$/mmc) embedded with a single Fe atom was used to simulate the catalyst according to a previous study[5]. Prior to generating the substrate structure, the lattice parameters of tridymite were optimised using the bulk structure to $a = 5.28$ Å and $c = 8.65$ Å. The substrate slab was built to contain a $4 \times 4$ surface SiO$_2$ unit repeated periodically. Three SiO$_2$ vertical layers were separated by a vacuum size of 10 Å. One surface Si atom and its two adjacent surface O atoms were replaced with an Fe and two C atoms, respectively. The top two layers of the slab were allowed to fully relax during geometry optimisations, whereas the bottom layer was kept fixed at the theoretical bulk-terminated geometry of tridymite.

The free energies of the gas and adsorbed species were calculated based on the ideal gas limit and harmonic limit, respectively, using the thermochemistry module implemented in the Atomic Simulation Environment package[21]. Vibrational frequencies were obtained using Hessian matrix calculations and normal mode analysis within a finite difference approximation. Transition-state geometries for the reactions were searched using the climbing-image nudged elastic band method and confirmed by frequency calculations to give one imaginary frequency.

Microkinetic modelling on the catalyst was performed under the mean-field approximation using the CatMAP software package[22]. The elemental reaction equations used for the microkinetic modelling are provided in the Supplementary Data 2. The contribution of each species, including transition-state species, to the overall reaction rate was estimated by the DRC[23],

$$\mathrm{DRC} = \frac{\partial \ln r}{\partial \left( \frac{-G_i}{k_\mathrm{B}T} \right)}, \qquad (1)$$

where $r$ is the overall reaction rate, $G_i$ is the Gibbs free energy for species $i$, $k_\mathrm{B}$ is Boltzmann's constant, and $T$ is temperature.

**Experimental**. A cristobalite silica lattice-confined 0.4 wt% Fe catalyst (Fe@CRS) was synthesised according to the previous report[9]. Synthesised nanocrystalline fayalite (Fe$_2$SiO$_4$) was mixed with commercial quartz (Kanto) at a ratio of 1:107. The solid mixture was then milled with three millimeter of zirconia balls at 250 rpm for 15 h without exposing O$_2$ for 15 h using a Pulverisette 7 premium line (Fritsch, Germany) apparatus. The resulting solid underwent the melt-fusing process at 1973 K for 6 h in air at a heating rate of 10 K min$^{-1}$ leading to a nonporous material (S$_{\mathrm{BET}} = 0.22$ m$^2$/g) with a high Fe$_3$O$_4$ dispersion.

Reaction measurements for non-oxidative methane coupling were carried out using a fixed-bed reactor system equipped with a quartz tube with an inner

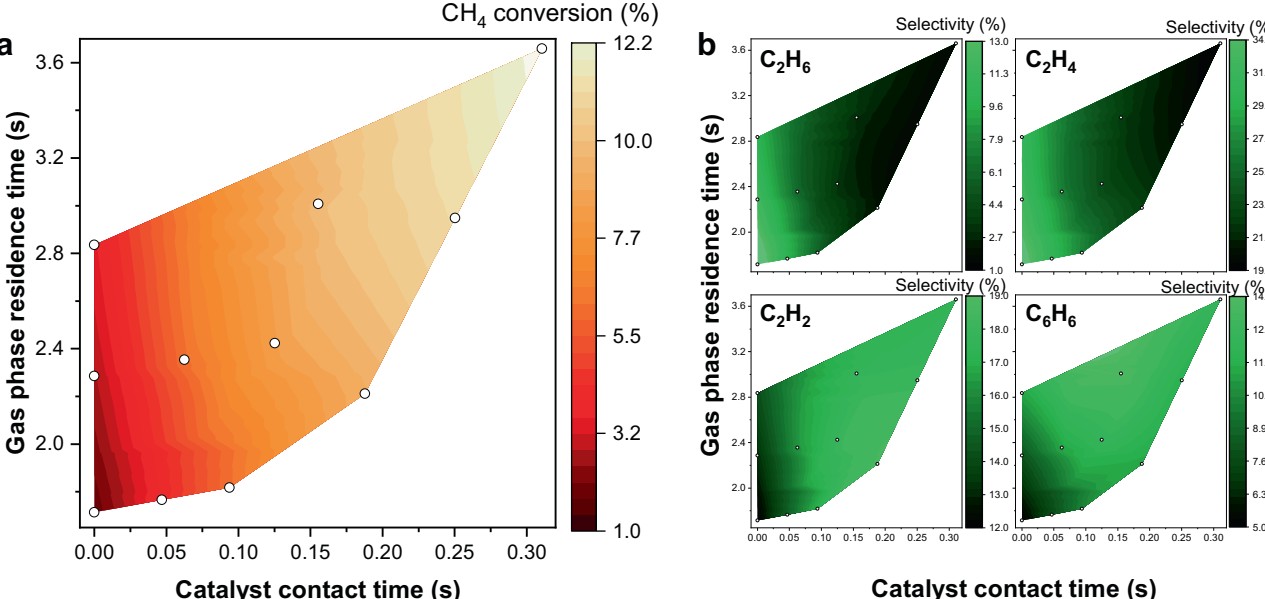

**Fig. 7 Experimental non-oxidative methane coupling results using Fe@CRS catalyst in a fixed-bed reactor. a** CH$_4$ conversion and **b** C$_2$H$_6$, C$_2$H$_4$, C$_2$H$_2$, and C$_6$H$_6$ selectivities as functions of catalyst contact time and gas-phase residence time. Data points are shown as white circles. Reaction conditions: $T = 1293$ K, $P = 0.1$ MPa, total residence time in the heating zone = 1.7–59.6 s, catalyst loading = 0–1.2 g.

diameter of 7 mm at 1293 K. The tubular reactor showed a uniform profile by bringing R-type thermocouples in direct contact with the outer surface. The reaction proceeded by feeding 90% $CH_4$/10% Ar mixed gas at the reaction temperature. The gas phase residence time (s) or catalyst contact time (s) was calculated by dividing the void space ($cm^3$) or solid volume ($cm^3$), respectively, by the feed flow rate ($cm^3$/s). The void space throughout which the gas-phase reaction proceeds was calculated by excluding the solid volume derived from the skeletal density of the catalyst from the reactor volume. The skeletal density and particle size of the Fe@CRS catalyst were 2.40 g/$cm^3$ and 425–850 μm, respectively[9]. The catalyst contact time and gas-phase residence time were varied by changing the flow rate and catalyst loading. The gas-hourly space velocity was varied from 0.1 to 0.6 s$^{-1}$, and the catalyst loading was varied from 0 to 12 g. We analysed gaseous reactants and products using an online GC (YL 6500GC, Younglin) equipped with one TCD connected with a ShinCarbon ST column (Restek, Catalogue No. 80486-800) and two flame ionisation detectors (FIDs) connected with an RT-Alumina BOND column (Restek, Catalogue No. 19756) and an Rtx-VMS column (Restek, Catalogue No. 49915). The methane conversion and product selectivity were determined according to the carbon balance based on the GC analysis.

X-ray absorption spectroscopy (XAS) measurements was conducted at beamline 8C (Nano XAFS) of the Pohang Accelerator Laboratory, operated at an energy of 3.0 GeV and a ring current of 70–100 mA. The obtained XAS results were analysed with the software package Athena[24].

## Data availability

All data supporting the findings are available within the Supplementary Information and Supplementary Data and from the corresponding author upon reasonable request.

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

## Acknowledgements

This research was supported by the C1 Gas Refinery Program through the National Research Foundation of Korea (NRF) funded by the Ministry of Science, ICT & Future Planning (NRF-2017M3D3A1A01037001). This research was further financially supported by the KRICT project (SI12011-30).

## Author contributions

S.K.K. and Y.T.K. guided the research and edited the manuscript. S.K.K., H.W.K., and J.S. performed the DFT calculations and microkinetic modelling. S.J.H. synthesised the catalyst and carried out XAS runs. S.W.L. and Y.T.K. performed the reaction tests. S.K.K. and Y.T.K. wrote the manuscript. All authors have reviewed the manuscript and have given approval to the final version.

## Competing interests

The authors declare no competing interests.
