## [Peer Review File · Communications Chemistry]

Reviewers' comments:

Reviewer #1 (Remarks to the Author):

Manuscript considers the high temperature non-oxidative coupling reactions of methane on single atom Fe/SiO₂ catalyst. Authors used density functional theory (DFT), thermodynamic and micro-kinetic mechanistic analysis to show, which proceeding reactions are predominant on catalyst or which in gas phase.

attached review file

Reviewer #2 (Remarks to the Author):

The manuscript described for mechanisms of the non-oxidative coupling of CH₄ over the Fe-single atom catalyst in SiO₂ based DFT and reactions.

The results of DFT are attractive; however, there is no essential and new data in the reaction test to prove the perfect conversion of CH₄ to higher hydrocarbons without carbon deposition.

I could not understand contribution of gas phase reactions. Many chemists have been observed the conversion of CH₄ to hydrocarbons and fine soot in the gas phase without catalyst at higher temperatures as similar conditions to your reaction conditions. why no contributions of the thermal gas phase reactions was in your reaction mechanisms?

I think that the in-situ XAFS data for the active site is essential for the publication.

The manuscript considers high temperature non-oxidative coupling of methane on single atom Fe/SiO₂ catalyst. The authors used DFT and microkinetic analysis to show which reactions are predominant on the catalyst and which in the gas phase. They show that the catalyst is mainly responsible for formation of CH₃ radicals and that acetylene is the major C₂ product that is formed on the catalyst surface. The study suggests that all other products observed during the reaction are a consequence of gas phase reactions. Computational findings were confirmed by the experiment. The manuscript is well written and the results are very interesting. The issues that need to be addressed are presented below.

1. “The previous study focused on the role of Fe-single atom catalyst that was capable of producing CH₃ radicals but did not participate in further C–C coupling of dehydrogenation.⁵”
– The non-oxidative methane activation, coupling, and conversion to ethane, ethylene, and hydrogen over Fe-based catalyst materials should be improved a bit, as there are other recent studies. All in all, the review of literature is modest, except is this is limited by the journal of choice.
2. At the beginning the reaction network of possible reactions on the surface of the catalyst at Fe active sites is presented. It is shown that transition-state scaling with the formation energy of the initial state is most appropriate for the 4 types of reactions considered. There is no clear connection with the rest of the manuscript. How does the information from this part (lines 102-109) relate to the rest of the study?
3. In the comparison of gas phase C-C coupling and catalytic C-C coupling the source of CH₃ radicals is not shown. How are these radicals generated? Why is this step not considered in the reaction mechanism and on internal energy and free energy diagrams?
4. Ethane adsorbs on the catalyst in the 1 -> 3.6e reaction. There should also be C₂H₆ in the adsorption arrows under the reaction network.
5. At microkinetic analysis TOF of product formation and CH₄ consumption on the catalyst are shown on Figure 3. Which reaction steps were considered here? All of the presented in the previous section (Fig. 1c)? This is not clear.

6. The major carbon product formed on the catalyst is acetylene. Is there any predominant pathway for acetylene formation on the catalyst?

7. In experimental catalytic tests the two reaction conditions are catalyst contact time and gas phase residence time. How were these two parameters varied independently? Was this achieved by varying the catalyst particle size and catalyst loading? Please provide details about particle size and WHSV.

8. There are also other products, besides C_2 , forming during the reaction on this catalyst as reported already in refs. 5 and 8. Is there a possibility that also other products are forming on the catalyst? The possibility of formation of other products on the catalyst should be discussed.

Thank you for the opportunity to improve this manuscript. We have addressed the remarks made by the reviewers as documented in this letter and in the attached document. The reviewer comments are included verbatim (blue), along with point-by-point responses (black). The revised text is also included (red).

Reviewer #1

The manuscript considers high temperature non-oxidative coupling of methane on single atom Fe/SiO₂ catalyst. The authors used DFT and microkinetic analysis to show which reactions are predominant on the catalyst and which in the gas phase. They show that the catalyst is mainly responsible for formation of CH₃ radicals and that acetylene is the major C₂ product that is formed on the catalyst surface. The study suggests that all other products observed during the reaction are a consequence of gas phase reactions. Computational findings were confirmed by the experiment. The manuscript is well written and the results are very interesting. The issues that need to be addressed are presented below.

1. “The previous study focused on the role of Fe-single atom catalyst that was capable of producing CH₃ radicals but did not participate in further C–C coupling of dehydrogenation.⁵”– The non-oxidative methane activation, coupling, and conversion to ethane, ethylene, and hydrogen over Fe-based catalyst materials should be improved a bit, as there are other recent studies. All in all, the review of literature is modest, except this is limited by the journal of choice.

Thank you for suggesting a more informative review of the literature. We searched for more research on Fe single-atom catalysts for this reaction and found a very recent study performed by van Bokhoven et al. (Chem.-Eur. J., 2020). They synthesized a new type of Fe single-atom catalyst and evaluated its catalytic performance in this reaction. Their Fe single-atom catalyst was highly active in the initial stage of the reaction, but it quickly became inactive due to the formation of large clusters. They also observed a change in the hydrocarbon distributions depending on the presence of Fe single sites. We summarized their report as below (new text in bold). Other studies adopting Fe single-atom catalysts have already been discussed in the Introduction and cited in the manuscript. (Reference #6–8)

The previous study focused on the role of Fe-single atom catalyst that was capable of producing CH₃ radicals but did not participate in further C–C coupling of dehydrogenation.⁵ **Furthermore, a recent study on a new type of Fe single-atom catalyst for non-oxidative methane coupling reported that the Fe single sites were only active in the initial reaction period, then rapidly lost their activity and resulted in varying hydrocarbon distributions.**⁶ Since the C–H bonds of ethylene and benzene are more readily activated than that of methane, it is thermodynamically challenging to inhibit coke deposition at such high temperatures.

2. At the beginning the reaction network of possible reactions on the surface of the catalyst at Fe active sites is presented. It is shown that transition-state scaling with the formation energy of the initial state is most appropriate for the 4 types of reactions considered. There is no clear connection with the rest of the manuscript. How does the information from this part (lines 102-109) relate to the rest of the study?

Thank you for pointing out this missing information. We randomly selected 42 transition states from the total of 108 presented in this study and expanded the initial-state scaling to predict the remaining ones, which significantly reduced the computational cost. To make this clearer, we edited the relevant text as below (new text in bold):

Among the 108 reactions presented here, 42 transition states were randomly selected, and their formation energies were scaled with initial-state and final-state formation energies in each category, as shown in Fig. 1d and Supplementary Fig. 1a, respectively.

...

We note that reactions in the transformation category exhibited weaker proportionality than other categories regardless of the scaling method because various types of bond cleavage and formation are combined differently for each individual transformation between surface hydrocarbon species. **The initial-state scaling derived as described above for each reaction category was used to predict the remaining transition-state energies for microkinetic modelling.**

3. In the comparison of gas phase C-C coupling and catalytic C-C coupling the source of CH₃ radicals is not shown. How are these radicals generated? Why is this step not considered in the reaction mechanism and on internal energy and free energy diagrams?

Since Fig. 2 compares the energetics of C–C coupling between the catalysis and gas-phase reactions, we did not show the CH₃ generation step. The CH₃ radicals were expected to be generated by CH₄ decomposition on the catalyst surface, which commonly takes place for both of the C–C coupling reactions. As mentioned in the manuscript, the free energies were compared under the assumption of $P_{\text{CH}_3}=0.01$ and $P_{\text{CH}_4}=1$. To clarify the purpose of Fig. 2, we edited the relevant text as below (new text in bold):

To investigate the role of the catalyst in C–C coupling, a typical pathway to produce ethylene on the surface was chosen in the reaction network, and its energetics were compared with that of the gas-phase reaction pathway. **We note that CH₃ radicals were assumed to be generated by the dehydrogenation of CH₄ on the catalyst surface, as illustrated in the reaction network (Fig. 1); this applies to both the catalytic and gas-phase C–C coupling pathways.** In the case of catalytic C–C formation (blue pathway in Fig. 2a), methane insertion to the surface (1.0 → 2.4b), dehydrogenation to produce surface radical species (2.4b → 2.3f), CH₃ addition to form surface C₂ species (2.3f → 3.6e), and remaining dehydrogenation followed by ethylene desorption (3.6e → ... → 1.0) were assumed to occur in turn.

4. Ethane adsorbs on the catalyst in the 1 → 3.6e reaction. There should also be C₂H₆ in the adsorption arrows under the reaction network.

Thank you for pointing out our mistake. We have revised the figure accordingly.

Fig. 1 | Surface structure of Fe single-atom catalyst and non-oxidative methane coupling reaction network.

5. At microkinetic analysis TOF of product formation and CH_4 consumption on the catalyst are shown on Figure 3. Which reaction steps were considered here? All of the presented in the previous section (Fig. 1c)? This is not clear.

In the microkinetic analysis, all reactions shown in Fig. 1c were involved. To make this point clearer, we edited the relevant text as below (new text in bold):

Fig. 3 shows how the CH_4 consumption rate and production rates change with reaction temperature and relative partial pressures of $\cdot\text{CH}_3$ and CH_4 . **We note that all the reactions presented in Fig. 1c were considered in this microkinetic analysis, except for gas-phase C–C coupling, to focus on the role of the Fe single-atom catalyst.** The CH_4 consumption rate monotonically increases with reaction temperature at a fixed $\cdot\text{CH}_3$ partial pressure.

6. The major carbon product formed on the catalyst is acetylene. Is there any predominant pathway for acetylene formation on the catalyst?

Thank you for the valuable comment. According to the reviewer's suggestion, we investigated the pathway for acetylene formation and found two competing pathways. We have added this information to the C_2 selectivity graph (Fig. 4) with a detailed explanation as below:

Fig. 4 | Product selectivity and dominant pathway for acetylene formation. **a**, CH_3 , H_2 , C_2H_4 , and C_2H_2 selectivities as functions of reaction temperature and partial pressure of CH_3 (P_{CH_3}). **b**, Two competing pathways to produce C_2H_2 in the reaction network, where CH_3 attaches to either C (intermediate 2.4a, pink lines) or Fe (intermediate 2.4b, yellow lines) on the surface. **c**, Free energy diagram of the two C_2H_2 formation pathways at $T = 1000$ K and $P_{\text{CH}_3} = 1$ bar.

Since the major C_2 product was identified as acetylene, the predominant pathway for acetylene formation was traced based on individual reaction rates. Two competitive reaction pathways were found as shown in Fig. 4b. Both pathways begin with a clean surface (1.0), but one proceeds by forming intermediate 2.4a, where CH_3 is attached to surface carbon (2.4a, pink lines in Figs. 4b and c), whereas the other proceeds by forming intermediate 2.4b, where CH_3 is attached to surface Fe (2.4b, yellow lines in Figs. 4b and c). The free energy diagram (Fig. 4c) shows that both pathways have comparable reaction barriers. However, microkinetic modelling of only these pathways shows that the pathway forming 2.4b (TOF = 1.3×10^{-4}) is slightly faster than the other path (TOF = 5.9×10^{-6}). This is presumably because the former proceeds via fewer reaction steps than the latter.

7. In experimental catalytic tests the two reaction conditions are catalyst contact time and gas phase residence time. How were these two parameters varied independently? Was this achieved by varying the catalyst particle size and catalyst loading? Please provide details about particle size and WHSV.

Thank you for letting us know that this important information was missing. We varied the catalyst loading and flow rate to change the catalyst contact time and gas-phase residence time. The catalyst particle size was 425–850 μm , the gas-hourly space velocity was varied from 0.1 to 0.6 s^{-1} , and the catalyst loading was varied from 0 to 1.2 g. We have added this information to the Experimental Methods section.

The skeletal density and particle size of the Fe@CRS catalyst were 2.40 g/cm^3 and 425–850 μm , respectively.⁹ The catalyst contact time and gas-phase residence time were varied by changing the flow rate and catalyst loading. The gas-hourly space velocity was varied from 0.1 to 0.6 s^{-1} , and the catalyst loading was varied from 0 to 12 g.

8. There are also other products, besides C_2 , forming during the reaction on this catalyst as reported already in refs. 5 and 8. Is there a possibility that also other products are forming on the catalyst? The possibility of formation of other products on the catalyst should be discussed.

As the reviewer pointed out, other products including C₃, C₄, C₅, naphthalene, toluene, and coke were produced during the experimental reactions. To focus on the formation of C₂ hydrocarbons—the main products—we did not discuss other hydrocarbon products. However, to provide more information to the readers, we have summarized the distributions of all the hydrocarbons in Supplementary Table 2 with the following additional text in the manuscript:

(In the Manuscript)

Other hydrocarbons, including C₃, C₄, and C₅, were also produced during the reaction, but their total amount was as low as 20–30% compared with that of C₂ hydrocarbons (see Supplementary Table 2). Coke deposition was also found to largely occur in some cases. It is noteworthy that the formation of C₃–C₅ hydrocarbons and coke increased with gas-phase residence time, indicating that the gas-phase radical reaction is considerably involved in these chain-growth reactions. Thus, further studies on the gas-phase reactions are necessary to aid in the design of catalysts and reactors.

(In the Supplementary Information)

Supplementary Table 2. Hydrocarbon distributions obtained using the Fe@CRS catalyst under various reaction conditions.

t _{gas} ^a (s)	t _{cat} ^b (s)	X ^c (%)	Selectivity (%)										
			C ₂ H ₆	C ₂ H ₄	C ₂ H ₂	C ₃	C ₄	C ₅	C ₆ H ₆	C ₇ H ₈	C ₁₀ H ₈	Alkyl benzenes	Coke
1.7	0.0	1.0	20.0	38.4	8.9	8.0	9.5	1.8	2.0	0.0	0.0	3.1	8.3
2.3	0.0	1.8	13.1	34.5	12.0	7.9	12.8	2.8	6.0	1.4	0.9	3.2	5.4
2.8	0.0	2.9	9.6	32.3	14.8	7.6	12.7	2.8	11.6	2.9	3.2	2.4	0.0
11.3	0.0	24.3	1.0	15.9	11.3	1.2	1.1	0.5	22.4	1.7	11.4	3.3	30.3
1.8	0.0	4.7	6.0	28.1	16.9	6.1	9.5	2.1	9.9	2.3	2.2	3.8	13.0
2.4	0.1	7.3	3.9	23.6	17.3	4.5	1.8	1.8	12.5	2.7	3.7	3.8	24.5
11.7	0.3	26.6	0.8	14.9	11.4	1.0	1.0	0.5	19.0	1.3	5.9	2.5	41.8
40.4	1.1	42.3	0.8	9.1	4.6	0.3	0.2	0.1	13.1	0.4	5.5	1.0	64.9
1.8	0.1	6.5	4.4	26.9	18.7	5.0	7.6	1.6	11.8	2.5	3.3	2.8	15.3
2.4	0.1	9.5	2.8	22.1	18.1	3.6	5.2	1.4	12.8	2.4	4.3	3.1	24.2
3.0	0.2	11.0	2.3	21.3	18.5	3.3	4.5	1.2	15.3	2.6	5.5	3.9	21.5
12.0	0.6	27.2	0.8	14.2	11.0	0.9	0.8	0.3	16.5	1.1	4.7	2.0	47.6
41.5	2.1	41.5	0.8	9.5	4.7	0.3	0.2	0.1	12.5	0.4	4.9	1.0	65.6
2.2	0.2	7.9	3.4	24.6	18.0	4.4	6.2	0.0	10.1	2.1	2.5	5.3	23.4
2.9	0.3	10.6	2.2	20.4	17.1	3.2	4.5	0.0	10.4	1.8	3.0	4.5	32.9
3.7	0.3	12.2	2.0	19.3	17.4	2.8	3.8	0.0	11.9	1.9	4.0	3.8	33.2
14.6	1.2	26.3	0.8	13.8	11.3	1.0	0.9	0.0	13.4	0.9	4.5	2.1	51.5
50.5	4.3	40.4	0.8	10.0	5.1	0.3	0.2	0.0	11.2	0.4	4.1	1.1	66.8

^a Gas-phase residence time

^b Catalyst contact time

^c Methane conversion

Reviewer #2

The manuscript described for mechanisms of the non-oxidative coupling of CH₄ over the Fe-single atom catalyst in SiO₂ based DFT and reactions. The results of DFT are attractive; however, there is no essential and new data in the reaction test to prove the perfect conversion of CH₄ to higher

hydrocarbons without carbon deposition.

I could not understand contribution of gas phase reactions. Many chemists have been observed the conversion of CH_4 to hydrocarbons and fine soot in the gas phase without catalyst at higher temperatures as similar conditions to your reaction conditions. Why no contributions of the thermal gas phase reactions was in your reaction mechanisms?

The motivation for this study is detailed in the Introduction. As the reviewer pointed out—and as we have described in the manuscript—non-oxidative methane coupling is highly complex, with both gas-phase and catalytic reactions occurring simultaneously. Thus, it has been challenging to identify their respective roles and to design reactors and catalysts accordingly. In the present study, we focused on catalytic reactions occurring on the surface and excluded complex gas-phase reactions. We believe this justification is well explained in the current manuscript.

I think that the in-situ XAFS data for the active site is essential for the publication.

Given the temperature conditions adopted for the experimental reaction (>1200 K), performing in-situ XAFS is unreasonable in practice. Instead, we have XAFS data for the Fe@CRS catalyst showing the phase changes before and after the reaction. We feel that this data is enough to elucidate how the catalyst structure changed during the reaction. We edited Fig. 6 and the relevant text as below:

In the X-ray absorption near-edge spectra, the spent Fe@CRS catalyst exhibited a white line at a position similar to those of the Fe foil and Fe_3C spectra, whereas the fresh catalyst showed an oxide-like spectrum. This indicates that the initial FeO_x clusters in the catalyst decomposed during the reaction and transformed into FeC_x .

Fig. 6 | X-ray absorption spectra of the fresh and spent $\text{Fe}@\text{CRS}$ catalysts and Fe foil, Fe_3C , and Fe_3O_4 reference samples. a, X-ray absorption near-edge spectra and b, extended X-ray absorption fine structures.

REVIEWERS' COMMENTS:

Reviewer #1 (Remarks to the Author):

Editorial note: The Reviewer has not provided any remarks to the authors.

Reviewer #2 (Remarks to the Author):

The manuscript is well revised and to be clear. I think that the revised manuscript is acceptable to the journal.